# Antibacterial Activity of Ikarugamycin against Intracellular *Staphylococcus aureus* in Bovine Mammary Epithelial Cells In Vitro Infection Model

**DOI:** 10.3390/biology10100958

**Published:** 2021-09-25

**Authors:** Shamsaldeen Ibrahim Saeed, Erkihun Aklilu, Khalid M. Mohammedsalih, Adewole A. Adekola, Ahmed Elmontaser Mergani, Maizan Mohamad, Nor Fadhilah Kamaruzzaman

**Affiliations:** 1Faculty of Veterinary Medicine, University Malaysia Kelantan, Pengkalan Chepa 16100, Kelantan, Malaysia; erkihun@umk.edu.my (E.A.); maizan.m@umk.edu.my (M.M.); 2Faculty of Veterinary Science, University of Nyala, P.O. Box 155, Nyala 63311, Sudan; marajan83@yahoo.com; 3Pathobiology and Population Science, Royal Veterinary College, Hawkshead Lane, Hatfield AL9 7TA, UK; aadekola18@rvc.ac.uk; 4Department of Microbiology, Faculty of Veterinary Medicine, University of Khartoum, Khartoum North 13314, Sudan; Ahmed.Mohamed@tiho-hannover.de; 5Institute for Biochemistry, University of Veterinary Medicine Hannover, Bünteweg 17, 30559 Hannover, Germany

**Keywords:** mastitis, antimicrobial resistance, intracellular bacteria, *S. aureus*, ikarugamycin

## Abstract

**Simple Summary:**

Antibiotics are widely used for the treatment and control of bovine mastitis. However, the treatment has only been partially effective, as the cure percentage only ranging from 10–30%. Infection by *Staphylococcus aureus* (*S. aureus*) is particularly difficult to treat due to the bacteria’s ability to enter and resides inside the host cells. Most antibiotics are ineffective against intracellular bacterial due to the poor penetration into host cells to achieve optimal intracellular bactericidal bioavailability levels. There is therefore, an increasing need to evaluate candidate active substances and develop novel antibiotics effective against intracellular persistence infection. In this study, we examine the potential antibacterial properties of ikarugamycin compound as an alternative drug candidate to be explored for treating persistent bovine mastitis caused by intracellular *S. aureus* using bovine mammary cell line as an in vitro infection model. We also assessed the potential cytotoxicity effect of ikarugamycin in the infection model. We found that, the ikarugamycin possessed intracellular killing activity against *S. aureus* within the mammary epithelial cell. This finding highlights the potential application of ikarugamycin as a novel antimicrobial for the treatment of *S. aureus* mastitis.

**Abstract:**

*Staphylococcus aureus* is an ubiquitous and versatile pathogen associated with a wide range of diseases. In animals, this bacterium is one of the causative agents of bovine mastitis, responsible for huge economic losses in the dairy industry. Besides the development of antibiotic resistance, the intracellular survival of *S. aureus* within udder cells has rendered many antibiotics ineffective, leading to therapeutic failure. Our study therefore aims to investigate the in vitro bactericidal activity of ikarugamycin (IKA) against intracellular *S. aureus* using a bovine mammary epithelial cells (Mac-T cells) infection model and determine the cytotoxic effect. Minimum inhibitory concentration (MIC) was used to determine the antibacterial activity of IKA, and Mac-T cells were infected with *S. aureus* using gentamicin protection assay. IKA intracellular antibacterial activity assays were used to determine the bactericidal activity of IKA against intracellular *S. aureus*. The cytotoxicity of IKA against Mac-T cells was evaluated using the resazurin assay. We showed that, *S. aureus* is susceptible to IKA with a MIC value of 0.6 μg/mL. IKA at 4 × MIC and 8 × MIC have bactericidal activity by reducing 3 and 5 logs_10_ CFU/mL of *S. aureus* in the first six-hour of treatment respectively. In addition, IKA demonstrated intracellular killing activity by killing 90% of intracellular *S. aureus* at 5 μg/mL. This level is comparatively lower than 9.2 μg/mL determined as the half-maximal inhibitory concentration (IC_50_) of IKA required to kill 50% of Mac-T cells, highlighting a lower concentration required for bactericidal effect compared to the cytotoxic effect. The study highlighted that importance of IKA as a potential antibiotic candidate to be explored for the in vivo efficacy in treating *S. aureus* mastitis.

## 1. Introduction

*Staphylococcus aureus* (*S. aureus*) is an ubiquitous bacterium and a major pathogenic agents associated with bovine mastitis in lactating dairy cows [1]. Mastitis in the dairy industry is arguably one of the most prevalent disease, leading to substantial economic loss due to the associated reduction in milk production, increased death and culling rates, and increased treatment costs [2,3]. The disease is also associated with profound welfare issues due to the associated morbidity and impact on livestock health [2,3]. Antibiotics administration is considered the most common strategy for treatment and control of bovine mastitis [4]. However, the use of antibiotics has become less effective due to the development of antibiotic resistance against common antibiotics used for mastitis treatment [4,5]. In addition, *S. aureus* can invade and survive inside bovine mammary epithelial cells [6], which creates an effective barrier limiting sufficient intracytoplasmic antibiotic bioavailability needed for effective bactericidal effect. This has been associated with poor treatment outcomes in *S. aureus* mastitis, where the percentage of cure using currently approved antibiotics is approximately 10–30% [7]. Intracellular *S. aureus* causing mastitis is less susceptible to common conventional antibacterial agents such as β-lactams, aminoglycosides, macrolides, and fluoroquinolones due to the inability to penetrate and accumulate in the mammalian cells [8]. The delivery of antibacterial into desired locations in the body is one of the main challenges for successful therapeutics as it needs to cross the host cell membranes either through diffusion or endocytosis [8]. The presence of intracellular *S. aureus*, therefore, provides privileged reservoirs from which re-infection can occur [9], resulting in the long-term disease course and recurrent infection [10]. The intracellular survival of *S. aureus* in mammary epithelial cells has thus been associated with the subclinical bovine mastitis and reinfection in treated dairy cows [11]. The facultative intracellular parasitism and biofilm production of *S. aureus*, therefore, protects them from host immune responses and the effect of antibiotics [12], and this poses huge treatment challenges for the global public and livestock health. Therefore, an urgent need to find novel antimicrobials with capabilities to enter, accumulate and retain within the mammalian cell to achieve therapeutic intracellular level.

Ikarugamycin (IKA) was first isolated as a natural active substance with antibacterial properties from *Streptomyces phaeochromogenes* subsp. *ikaruganensis* [13] and belongs to the polycyclic tetramate macrolactams (PTMs) class [13]. Structurally, these classes are similar, containing macrocyclic lactam ring, tetramic acid ring, and a variable set of a carbocyclic rings system. IKA possess 5-6-5 carbocyclic rings, while other PTMs harboring 5-5-6 or 5-5-5 carbocyclic rings [14]. IKA and other PTMs have been reported to possess various biological properties such as antimicrobial [15,16], antiprotozoal [17], antitumor [18], immune regulation [19] as well cytotoxic properties [20]. IKA has also been widely used as an inhibitor for the clathrin-mediated endocytosis uptake pathway in the mammalian cells [20]. However, neither its intracellular antibacterial potency nor its cytotoxic properties towards bovine mammary epithelial cells (Mac-T cell) have been examined. Our present study, therefore, aims to investigate the in vitro antibacterial activities of IKA against intracellular *S. aureus* in Mac-T cells and the cytotoxicity of the compound.

## 2. Materials and Methods

### 2.1. Reagent

IKA purchased from the TOKU-E company (Batch No: 36531-78-9, Toku-E, Tokyo, Japan) was used for this study. IKA was prepared as a 1 mg/mL stock solution dissolved in dimethyl sulfoxide (DMSO) (Sigma-Aldrich Chemie GmbH, Steinheim, Germany) and stored at −80 °C. The stock solution was further diluted before use for our experiments in phosphate-buffered saline (PBS) to achieve a final DMSO concentration less than 0.01% (Figure 1).

### 2.2. Bacterial Strains and Culture Conditions

Two *S. aureus* strains (F51B and 15 AL) were used for this study. *S. aureus* F51B was isolated from cases of subclinical mastitis on dairy farms on the east coast of Malaysia while the *S. aureus* strain 15 AL strain was obtained from Dr. Shan Goh (Royal Veterinary College, London, UK). All these isolates have previously been confirmed as multi-drug resistance and capable of intracellular localization in mammary epithelial cells. Isolates selected for our were sensitive to chloramphenicol, ciprofloxacin, kanamycin, sulphonamide and gentamicin. The isolates were maintained in −80 °C freezer stock. Before the experiment, the bacteria were cultured in Mannitol Salt Agar (MSA) and subsequently grown in Mueller Hinton Broth (MHB) (Oxoid, New Hampshire, UK) at 37 °C in a shaking incubator at 200 rpm for 24 h.

### 2.3. Determination of Minimum Inhibitory Concentrations (MIC) and Minimum Bactericidal Concentration (MBC)

MIC was performed according to CLSI, 2016. Briefly, the overnight culture of *S. aureus* was diluted in MHB to achieved 5 × 10^5^ CFU/mL. The diluted suspension was added into a 96-well microplate containing a range of IKA concentrations in 200 μL final volume, followed by incubation at 37 °C for 18 h. The MIC is defined as the lowest concentration of IKA that inhibited the growth of bacteria. For our study, the MBC was determined as the lowest IKA concentration required to reduce 99.9% of CFU/mL on the agar plates. Control positive and negative were included and each experiment was performed in triplicate [21].

### 2.4. Time-Kill Assays

Time-kill kinetics assays were performed according to (CLSI, 2016) to determine the bactericidal properties of IKA. Briefly, overnight culture of, *S. aureus* was diluted in tryptic soy broth (TSB). Following that, bacterial was added into tubes containing a designated concentration of IKA (1 × MIC, 4 × MIC, and 8 × MIC) to obtain the final bacterial concentration of 1 × 10^5^ CFU/mL. The untreated bacteria were included as control. Following that, the tubes were incubated at 37 °C in a shaking incubator and 100 μL of suspension was taken out at various times (3, 6, 12, 24 h) followed by serial dilution, and plated on nutrient agar. The plates were incubated at 37 °C for 18 h followed by colony counting to determine the time-killing kinetics. The data were plotted with time against the logarithm of the viable count. Each experiment was performed in triplicate.

### 2.5. Bovine Mammary Epithelial Cell Cultures

Bovine mammary epithelial (MAC-T) cells were obtained from Dr. Amanda Gibson, Royal Veterinary College, London, UK, and used as a host for the in vitro intracellular infection model. The cells were grown in cell culture flask in Dulbecco’s modified Eagle’s medium (DMEM) (Sigma-Aldrich, St. Louis, MO, USA) supplemented with 10% fetal bovine serum (FBS) (Sigma-Aldrich, St. Louis, MO, USA), and 5% penicillin-streptomycin (Sigma-Aldrich, St. Louis, MO, USA). The cells were incubated in 5% CO_2_ at 37 °C.

### 2.6. Intracellular Invasion Assay

In vitro infection of MAC-T cells was performed using gentamicin protection assay as previously described [7]. As shown in Figure 2, aliquots of *S. aureus* diluted to 1 × 10^7^ to CFU/mL were co-incubated with host cells for 3 h for intracellular internalization. Following that, gentamicin (200 mg/L) was added for another 3 h to kill extracellular bacteria. Finally, the infected Mac-T cell was lysed using 0.5% Triton X-100 and the lysed cells were serial diluted and plated on nutrient agar to obtain the CFU/mL of intracellular bacteria [7].

### 2.7. Intracellular Antibacterial Activity Assay

The intracellular antibacterial activities of IKA were performed as previously described [7]. Briefly, MAC-T cells were invaded by *S. aureus,* followed by incubation with gentamicin to kill the extracellular bacteria. Following gentamicin exposure, host cells were rinsed with PBS. Subsequently, the wells containing infected cells were treated with IKA at 1 × MIC, 4 × MIC, 8 × MIC, and cells without IKA treated were used as control. Plates were incubated for 3 h to kill the intracellular bacteria. Afterward, IKA was removed, and the cells were lysed after rinsing. Lysed cells were serially diluted and plated on nutrient agar to quantify the remaining survival of bacteria [6,7].

### 2.8. Cytotoxicity Assay

The cytotoxicity of IKA toward Mac-T cells was assessed using resazurin assay [23]. Resazurin is a cell-permeable non-fluorescent dye that can be used to monitor the number of viable cells. The blue colored resazurin will be reduced and change into pink color resorufin by the active and viable mammalian cells. Briefly, Mac-T cells (4 × 10^4^ cells/well) were seeded in 96-well plates for 48 h, followed by co-incubation with increasing concentration of IKA up to 10 μL/mL at 37 °C for 3 h. Control positive and negative were included and each experiments were performed in triplicate. Resazurin dye at 44 μM final concentration was added to each well, and then the plate was incubated for 48 h. Finally, the optical density (OD) was measured using POLARstar Omega microplate reader (BMG, Labtech, Germany) at 550 and 630 nm. Survival curves were plotted, and the IC_50_ (inhibitory concentration 50%) for IKA was calculated using GraphPad Prism version 8.0 (San Diego, CA, USA).

### 2.9. Statistical Analysis

Data were analyzed by using GraphPad Prism 8 (San Diego, CA, USA). Statistical analysis was performed using a one-way analysis of variance (ANOVA) with Tukey test. Data are presented as means ± standard deviation (SD). The level of significance was accepted as *p* ≤ 0.05. All experiments were performed at least three times.

## 3. Results

### 3.1. Minimum Inhibitory Concentration and Minimum Bactericidal Concentration of Ikarugammycin

MIC was used to evaluate the antibacterial susceptibility of IKA towards *S. aureus*. The MIC value for IKA was 0.6 and 5 μg/mL MBC value.

### 3.2. Time-Kills Assay

The results of the time curves assay show that, IKA has bactericidal activity against *S. aureus* (reducing more than 3 log_10_) at 2.5 and 5 μg/mL towards the tested bacteria following 6 h of exposure (Figure 3).

### 3.3. Intracelluar Infection of Mac-T Cells by S. aureus

To test the antibacterial activity of IKA against intracellular *S. aureus*, and in vitro infection modeling of Mac-T cells by *S. aureus* were done using the gentamicin protection assay. Two isolates were confirmed to invade Mac-T cells as indicated by the increase of survival of *S. aureus* after gentamicin exposure and lysis of Mac-T cells. Lysis of Mac-T cells released approximately 10^5^ CFU/mL of *S. aureus* 15 AL and 10^4^ CFU/mL of F3 3D (Figure 4).

### 3.4. Bactericidal Activities of Ikarugammycin against Intracellular Staphylococcus aureus

To determine the antibacterial activity of IKA against intracellular *S. aureus*, Mac-T cells were infected with *S. aureus* using the gentamicin protection assay. Infected MAC-T cells were treated with different concentrations of IKA (0.6, 2.5, and 5 μg/mL) for 3 h. Following that, IKA was removed, and the cells were lysed to calculate the CFU of surviving intracellular bacteria. The results indicated that, IKA at 5 μg/mL killed between 85 to 90% of intracellular *S. aureus* (Figure 5).

### 3.5. Cytotoxicity Assay

Cytotoxicity test was performed using resazurin reduction assay. Our results highlighted that, the concentration of IKA that inhibited 50% of MAC-T cells (IC_50_) was 9.2 μg/mL; which is higher than the concentration required to kill 90% of intracellular *S. aureus* in Mac-T cells Figure 6.

## 4. Discussion

*Staphylococcus aureus* causing mastitis is a challenging disease in dairy animal and proven to be difficult to combat using conventional antimicrobials due to the development of antimicrobial resistance, and intracellular localization of the bacteria inside the host cell. Here in this study, we demonstrated the antibacterial activities of IKA against intracellular *S. aureus* and cytotoxic properties toward bovine mammary epithelial cells (Mac-T cells). IKA was effective against the extracellular and intracellular *S. aureus* and demonstrated low cytotoxicity towards bovine mammary cells.

IKA is a natural product belonging to the polycyclic tetramate macrolactams (PTMs) family [13]. The antibacterial activities of IKA have been hypothesized to be partly due to the interference of the tetramic acid groups with the bacterial proton gradient and membrane potential. This interference could eliminate the cellular proton motive force (PMF) (which plays a crucial role in ATP synthesis and solute transport across the cell) through depletion of the transmembrane proton gradient (ΔpH) which results to cell death [24]. In addition, macrocyclic lactam ring in IKA could bind to the D-alanyl-D-alanine of bacterial cell wall, thus inhibit the peptidoglycan biosynthesis and disrupt the cell membrane integrity [24,25].

The ability of IKA to kill intracellular activity was probably attributed to several factors. IKA was reported to enter the mammalian cells, and subsequently bind to the DNA [26,27,28]. Therefore, there are high chances for interaction between the compound and intracellular *S. aureus* within Mac-T cells. Additionally, IKA is a known compound to demonstrate antitumor activities, attributed to the ability of the compound to induce autophagy of the cells. Cells exposed to the IKA showed increase in the concentration of cytosolic Ca^2+,^ followed by activation CaMKKβ and AMPK pathway, and ultimately in promotion of autophagy [26]. Autophagy is a cellular processes that target intracellular components including intracellular bacteria [29,30].

Our study also found that the IC_50_ of IKA (the concentration required to inhibit 50% of MAC-T cells) was 9.2 μg/mL which was comparatively higher than the concentration required to kill 90% of intracellular *S. aureus* in MAC-T cells. This data showed that, IKA is effective as an antimicrobial agent. Several studies have reported the cytotoxic properties of IKA, and this has been explored for its anti-tumor property [20]. No data are, however, available on the cytotoxicity using MAC-T cells and the precise mechanism of toxicity. Several other studies measured IKA toxicity towards the following cell line, HL-60 human promyelocytic leukemia cells [31], pancreatic cancer cells [18], human monocytic cells [19], mouse macrophage 5774 [32]. IKA toxicity levels varied and highly dependent on the time of exposure and dose of the compound [31]. Nevertheless, in our study, Mac-T cells appeared to have tolerance towards IKA with cell viability were only affected when cells were exposed to IKA at a concentration higher than the concentration required to kill intracellular *S. aureus* in the host cells.

## 5. Conclusions

The present study demonstrated that, IKA has antibacterial activity against intracellular *S. aureus* in in vitro models in Mac-T cells with host cells tolerance of IKA at concentrations higher than the required concentration to kill the intracellular *S. aureus*. This highlights the potential importance of IKA as an alternative antimicrobial candidate to be explored in the treatment of persistent bovine mastitis caused by *S. aureus*. Further in vivo studies are needed to evaluate IKA potency and toxicity to further understand the potential application of this compound for bovine mastitis.

## Figures and Tables

**Figure 1 biology-10-00958-f001:**
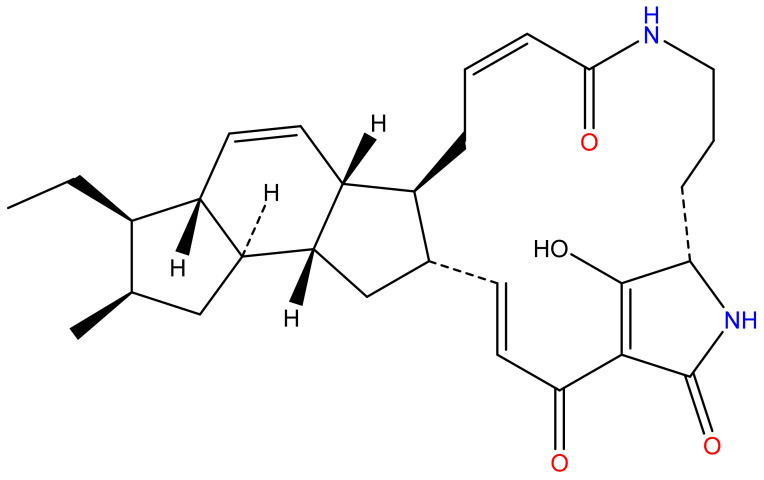
Structure of IKA, which consists of macrolactam ring, tetramic acids, and carbocycles [16]. Image was created using ChemDraw software 14.0 (PerkinElmer, Inc., Akron, OH, USA).

**Figure 2 biology-10-00958-f002:**
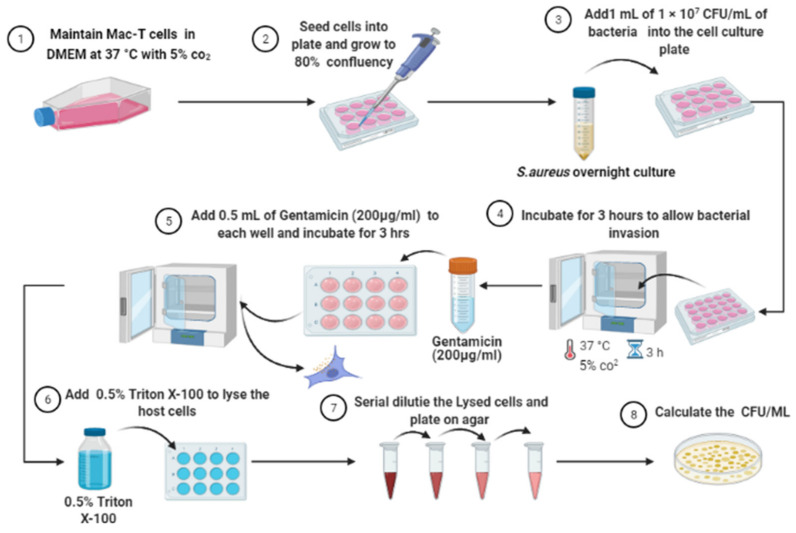
Flowchart of intracellular infection of bovine mammary epithelial cells (Mac-T cells) by *S. aureus* using a gentamicin protection assay [22]. The image was created using BioRender.com.2021 (Toronto, ON, Canada).

**Figure 3 biology-10-00958-f003:**
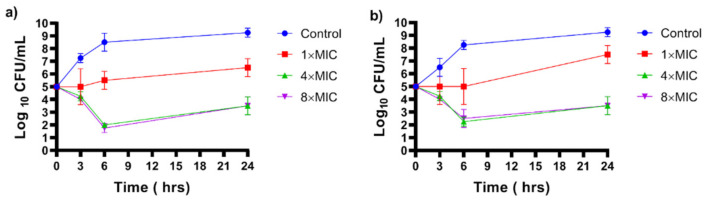
Time–kill curves of *S. aureus* 15AL (**a**) and *S. aureus* F53 B (**b**) following exposure with IKA for 24 h at 37 °C. For each time point, an aliquot of bacteria was taken out, serially diluted and plated for colony counting. The experiments were performed in triplicates and each value represents the mean ± standard deviation of log CFU/mL of bacteria.

**Figure 4 biology-10-00958-f004:**
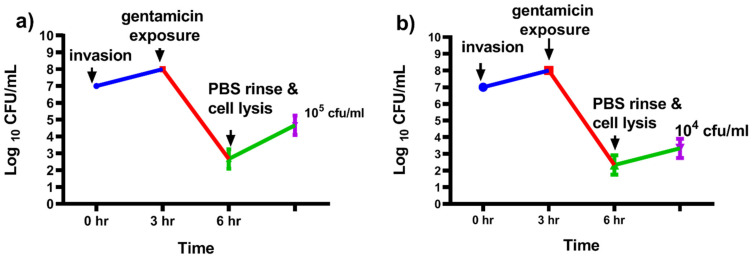
Intracellular survival of *S. aureus* in Mac-T cells. Following gentamicin exposure, lysis of Mac-T cells released approximately 10^5^ CFU/mL of *S. aureus* (**a**) 15 AL and (**b**) 10^4^ CFU/mL of F3 3D. Error bars represent the standard deviation (SD) of triplicates.

**Figure 5 biology-10-00958-f005:**
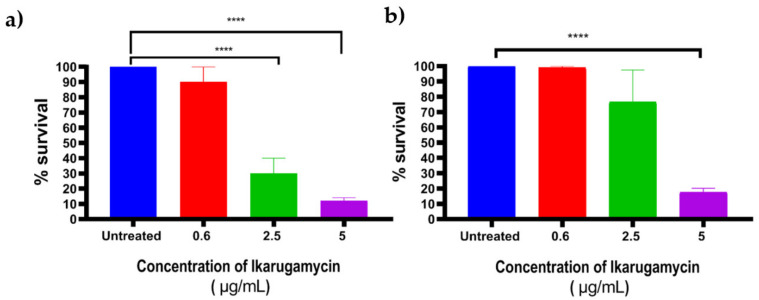
Antibacterial activity of IKA against intracellular *S. aureus* (**a**) 15 AL and (**b**) F53 B. Error bars represent the standard deviation (SD) of triplicates. **** *p* ≤ 0.0001.

**Figure 6 biology-10-00958-f006:**
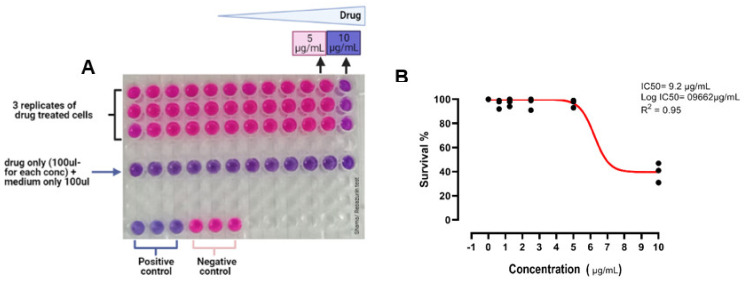
(**A**) 96-well plate showing resazurin reduction assay after 48 h; (**B**) dose-response curve showing the IC50 of IKA toward MAC-T cells.

## Data Availability

Not applicable.

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
