# Peer review of "Antibacterial Activity of Ikarugamycin against Intracellular Staphylococcus aureus in Bovine Mammary Epithelial Cells In Vitro Infection Model"

_biology, 2021, doi:10.3390/biology10100958_

Round 1
Reviewer 1 Report
The revised manuscript improved and many suggestions were incorporated; however, some comments were not adequately addressed and there is still room for improvement in the current manuscript.
- The English language was not fully revised since many several grammatic mistakes and typos are still present throughout the manuscript. Please revise.
- The main concern is still in the time-kill assay procedure. S. aureus were diluted to a concentration of 5x10^5 cfu/ml and then added to the tubes containing IKA? This is not correct because the final concentration of the inoculum would be lower than 5x10^5. If so, this experiment must be performed again with the final concentration of the inoculum being 5x10^5. This comment was included in the previous review but not addressed at all.
Author Response
The revised manuscript improved and many suggestions were incorporated; however, some comments were not adequately addressed and there is still room for improvement in the current manuscript.
- The English language was not fully revised since many several grammatic mistakes and typos are still present throughout the manuscript. Please revise.
Thank you for the comment. We have proofread the manuscript and made appropriate changes to the errors.
- The main concern is still in the time-kill assay procedure. S.aureus were diluted to a concentration of 5x10^5 cfu/ml and then added to the tubes containing IKA? This is not correct because the final concentration of the inoculum would be lower than 5x10^5. If so, this experiment must be performed again with the final concentration of the inoculum being 5x10^5. This comment was included in the previous review but not addressed at all.
Thank you for the comment. We have revised and rechecked the experiment. Following adding the bacterial culture to the drug, the final inoculum changed to 1x10^5 CFU/ML
Reviewer 2 Report
Thank you for the authors' point-by-point responses to your comments. But the manuscript still had huge limitation in my opinion. First of all, in vitro test has great limitation which cannot provide enough evidence that the IKA compound against intracellular S. aureus. So, additional animal testing is recommended. Secondly, there are a large number of grammar and writing errors in the manuscript, please further modify and improve.
Author Response
Thank you for the authors' point-by-point responses to your comments. But the manuscript still had huge limitation in my opinion. First of all, in vitro test has great limitation which cannot provide enough evidence that the IKA compound against intracellular S. aureus. So, additional animal testing is recommended. Secondly, there are a large number of grammar and writing errors in the manuscript, please further modify and improve.
Thank you for the comment. We have proofread the manuscript and corrected errors.
We understand the importance of an in vivo study for the validation of the pharmacological potency of IKA. However, our study was focused on the in vitro performance of IKA as a justification for further in vivo studies to explore the pharmacological and safety in bovine mastitis-infected animals.
The limitation has been included in the conclusion section of our manuscript along with changes to the title of our article to reflect the in vitro focus of our study.
Reviewer 3 Report
The authors have provided a corrected version that addressed most of the concerns raised previously. This is really good work and the authors must be congratulated for such an improvement.
However, the resubmitted manuscript still fails to address an important concern regarding the clinical importance of this work.
There is no mention anywhere regarding pharmaceutical forms available, the authors do not touch at all upon matters of administration, for example injection or intramammary infusion. Also, the authors do not justify their work, why did they use this antibiotic? There are many other active substances than can be used in similar studies.
All in all, I am not happy with this manuscript.
Before possible acceptance, the above issues must be addressed. Otherwise, rejection will be the only option.
Author Response
The authors have provided a corrected version that addressed most of the concerns raised previously. This is really good work and the authors must be congratulated for such an improvement.
However, the resubmitted manuscript still fails to address an important concern regarding the clinical importance of this work.
There is no mention anywhere regarding pharmaceutical forms available, the authors do not touch at all upon matters of administration, for example, injection or intramammary infusion. Also, the authors do not justify their work, why did they use this antibiotic? There are many other active substances that can be used in similar studies.
All in all, I am not happy with this manuscript.
Before possible acceptance, the above issues must be addressed. Otherwise, rejection will be the only option.
Thank you for the comment. The antibiotic is currently not licensed for animal usage. The work was initially part of existing research exploring the use of ikaguramycin as an endocytic inhibitor. We observed an unanticipated activity of IKA towards intracellular S. aureus and therefore decided to further expand the study to look into the toxicity of the compound towards the Mac-T cells. The justification for the consideration of this active substance was therefore due to its ability to achieve substantial intracellular bioaccumulation making it an ideal candidate for treating bovine mastitis. This current study, therefore, focused on the in vitro bactericidal performance of IKA as a justification for further in vivo studies to explore the pharmacological potential and safety in treating bovine mastitis. This focus has been reflected in the revised title of our article. The pharmaceutical form and route of administration would therefore be dependent on the outcomes of such in vivo study. Unfortunately, such in vivo study is outside the scope of our work due to limited available funds.
Round 2
Reviewer 1 Report
The authors have addressed all commentaries.Reviewer 2 Report
The author has been sufficiently revised the manuscript and I consider it suitable for publication
Reviewer 3 Report
I read the answer of the authors and I leave it to the editor to decide for final acceptance in relation to this answer.